# Precise Printing of Microfiber Scaffold with Gelatin Methacryloyl (GelMA)/Polyethylene Oxide (PEO) Bioink

**DOI:** 10.3390/bioengineering10020130

**Published:** 2023-01-18

**Authors:** Haibing Li, Ruijian Zhou, Qiang Shu, Mingjun Xie, Yong He

**Affiliations:** 1Department of Paediatric Orthopaedics, The Children’s Hospital Zhejiang University School of Medicine, Hangzhou 310052, China; 2Department of Plastic and Reconstructive Surgery, Zhejiang Provincial People’s Hospital, Hangzhou Medical College, Hangzhou 310014, China; 3Key Laboratory of Fluid Power and Mechatronic Systems, School of Mechanical Engineering, Zhejiang University, Hangzhou 310027, China; 4Key Laboratory of Materials Processing and Mold, Zhengzhou University, Zhengzhou 450002, China; 5Key Laboratory of 3D Printing Process and Equipment of Zhejiang Province, College of Mechanical Engineering, Zhejiang University, Hangzhou 310027, China; 6Cancer Center, Zhejiang University, Hangzhou 310058, China

**Keywords:** electrospinning, microfiber, hydrogel, 3D bioprinting, tissue engineering

## Abstract

Gelatin methacryloyl scaffolds with microscale fiber structures own great significance because they can effectively mimic the extracellular matrix environment. Compared with extruding bioprinting, electrospinning technology is more suitable for establishing accurate hydrogel microfibers. However, electrospinning accurate gelatin methacryloyl microfiber remains a big challenge restricted by its bad spinnability. In this paper, polyethylene oxide, which owns promising spinnability, is added into gelatin methacryloyl hydrogel precursor to improve the spinnability of gelatin methacryloyl bioink. A three-dimensional motion platform for electrospinning is designed and built and the spinning process of microfibers under far-electric-field and near-electric-field conditions is systematically studied, respectively. As a result, scaffolds consisted of unordered and ordered microfibers are successfully fabricated under far-electric-field and near-electric field, respectively. In vitro culture experiments of human umbilical vein endothelial cells are carried out using the prepared gelatin methacryloyl microfiber scaffolds. The results show that the cells can easily attach to the microfibers and grow well. Moreover, the gelatin methacryloyl/ polyethylene oxide microfiber scaffold was directly spun on the polycaprolactone mesh scaffold printed by fused modeling printing method. The results showed that the macroscopic ordered and microscopic disordered microfiber scaffold could be successfully established, which could lead to directed cell growth. We believe that this method can effectively solve the problem of hydrogel spinnability and be a powerful tool for various biomedical engineering methods in the future.

## 1. Introduction

As an emerging interdisciplinary topic, tissue engineering involves mechanical and biological material fields and has broad development prospects in multiple directions [1,2,3,4,5]. Typical clinical applications of tissue engineering scaffolds mainly include three steps: (i) culturing and expanding the target functional cells in vitro; (ii) attaching cells or growth factors to the scaffold and transplanting into the body; and (iii) new tissue gradually replacing the scaffold to achieve the purpose of wound repair. Thus, for tissue repair applications, good tissue engineering scaffolds need to have a certain amount of mechanical strength, good biocompatibility and degradability. At the microscopic level, the main functional component of the natural extracellular matrix (ECM) is fibrin. Specifically, fibrin provides the basis for the orderly integration of a variety of cells, and its spatial structure characteristics such as fiber morphology and arrangement can effectively support cell growth, migration, proliferation and differentiation activities. In addition, microfibers have the characteristics of a large specific surface area and high porosity, which are ideal structures for constructing biomimetic extracellular matrices. Therefore, the preparation of microfiber structures with excellent biological properties through advanced manufacturing methods is the key to breakthroughs in tissue repair.

The traditional methods for preparing microfibers mainly include self-assembly method, template synthesis method, phase separation technology and lithography technology, etc. However, these methods always require high costs and limited material types. Electrospinning technology [6,7,8,9] has aroused widespread interest in both academia and industry due to the advantages of simple equipment, the wide range of optional materials and simple process methods, etc. Moreover, the prepared fiber uniformity and continuity are high, which can better simulate the 3D collagen fiber network structure of the natural extracellular matrix. The scaffold electrospinning process based on some polymer materials, such as polylactic acid (PLA) and polycaprolactone (PCL), etc. [10,11,12], has been explored and analyzed by some researchers and put into a series of biomedical applications.

However, the microfiber scaffolds based on the above materials are not always promising due to the low biocompatibility of the normal polymer materials. In recent years, hydrogels have been widely applied due to their good printability and biocompatibility [13,14,15,16,17,18,19]. Among them, gelatin methacryloyl (GelMA), which is modified from gelatin [20,21], has become more and more popular due to its great biocompatibility for cellular functionalization and rapid photo-cross-linking capability, which has been introduced into a series of biomedical applications [21,22,23,24,25] accompanied with the development of 3D bioprinting [26,27,28,29,30,31], such as disease models, drug delivery and tissue repair. Therefore, combining GelMA hydrogel and electrospinning technology will potentially establish promising tissue engineering scaffolds with both microscale morphology and better biocompatibility. However, the spinnability of GelMA was too bad due to its low viscosity [32,33,34,35]. Therefore, electrospinning GelMA microfibers, especially the scaffolds with ordered microfiber patterns, remains a big challenge.

Here, polyethylene oxide (PEO), which owns promising spinnability, is added into GelMA hydrogel precursor to improve the spinnability of hydrogel bioink. At room temperature, PEO is miscible with water in any proportion. The aqueous PEO solution with a relative molecular weight of about 10^4^ has spinning performance at 1% (*w*/*v*), and a non-viscous elastic gelation state at 2% (*w*/*v*). A three-dimensional (3D) motion platform for electrospinning is designed and built (Figure 1A,B) and the spinning process of microfibers under far-electric-field (FEF) and near-electric-field (NEF) conditions are systematically studied, respectively. As a result, scaffolds consisted of unordered and ordered microfibers are successfully fabricated under far-electric-field and near-electric-field, respectively. In vitro culture experiments of human umbilical vein endothelial cells (HUVECs) are carried out using the prepared GelMA microfiber scaffolds. The results show that the cells can easily attach to the microfibers and grow well. Moreover, the GelMA/PEO microfiber scaffold was directly spun on the PCL mesh scaffold printed by fused modeling printing (FDM) method, which could lead to directed cell growth. We believe that this method can effectively solve the problem of hydrogel spinnability and provide a powerful tool for various biomedical engineering in the future.

## 2. Materials and Methods

### 2.1. Materials and Device

GelMA powder and photoinitiator LAP were purchased from Suzhou Intelligent Manufacturing Research Institution (EFL-GM-30, Suzhou, China). PEO powder was purchased from Sigma-Aldrich Co. Ltd. (1000 kDA, St. Louis, Mo, USA). Dulbecco’s modified eagle medium (DMEM) was purchased from Gibco Co. Ltd. (Carlsbad, CA, USA). Type II collagenase was purchased from BioSharp Co. Ltd. (Hefei, China). Phosphate buffer solution (PBS) was purchased from Tangpu Co., Ltd. (Hangzhou, China). Deionized water was purchased from Diena Co., Ltd. (Shanghai, China). Polycaprolactone (PCL) was purchased from Perstorp Co., Ltd. (80kDa, Shanghai, China). Super clean bench was purchased from Suzhou Purification Equipment Co., Ltd. (SW-CJ-2G, Suzhou, China). High-speed camera was purchased from PCO Imaging Co., Ltd. (Dimax S1, Munich, German). Fused deposition modeling (FDM) printer was purchased from Suzhou Intelligent Manufacturing Research Institution (EFL-BP, Suzhou, China). Capital microscope was purchased from Feica Co., Ltd. (DMi1, Wetzlar, German). Fourier infrared spectroscopy analyzer was purchased from Thermo Electron Instrument Co., Ltd. (Nicolet5700, Waltham, Ma, USA). Vacuum freeze dryer was purchased from Ningbo Xinzhi Freeze Dryer Instrument Co., Ltd. (SCIENTZ-100F, Ningbo, China). Electronic balances were purchased from METTLER TOLEDO Co., Ltd. (ME104E, Zurich, Switzerland). Magnetic stirrer was purchased from Xichengxinrui Instrument Co., Ltd. (HJ-6B, Changzhou, China). Vortex mixer was purchased from Kylin-Bell Lab Instruments Co., Ltd. (QL-861, Haimen, China). Syringes were purchased from Shanghai Kindly Medical Instruments. Co., Led. (1 mL, Shanghai, China). Syringe filters were purchased from Millex-GP Co., Led. (0.22 μm, Boston, MA, USA).

### 2.2. Electrospinning Platform Building

The hydrogel electrospinning platform consisted of six modules, namely upper computer, numerical control system, three-axis motion system, high-voltage DC power supply, controllable syringe pump and receiving plate. High-voltage electric field was generated between the nozzle tip and the receiving plate. In order to meet the high-precision control requirement of the electrospinning process, especially the NEF direct writing, the three-axis motion stage in the form of XYZ was chosen, the two key XY degrees of freedom of which were controlled through an optical precision translation stand. The bioink supply part was realized by controlling the stepper motor of the syringe pump by the syringe pump controller. The overall control system of the platform was similar to that of a 3D printer, which could meet the needs of NEF direct writing.

### 2.3. GelMA/PEO Bioink Preparation

Pure GelMA precursor was prepared by dissolving the freeze-dried GelMA (EFL-GM-30, 10% (*w*/*v*)) and lithium phenyl-2, 4, 6-trimethylbenzoylphosphinate (LAP, 0.5% (*w*/*v*)) in deionized water and being shaken in a vortex mixer for 5 min to mix evenly. The solution was then filtered through a 0.22 μm filter. Then, an appropriate amount of PEO powder (sterilized under UV light for 1 h) and the prepared GelMA precursor were simultaneously added into the brown bottle and stirred on a magnetic stirrer at a speed of 500 r/min for 12 h to mix evenly. The prepared GelMA/PEO bioink should be kept in a dark environment in the 4 °C refrigerator.

### 2.4. Hydrogel Degradation Test

Five parts of GelMA mass volume concentration of 10% and PEO mass volume concentration of 0%, 1%, 2%, 3% and 4% were prepared, and cylindrical samples of consistent size were prepared under the condition of 3 cm blue light irradiation for 20 s. Then, 20 mg type II collagenase was added to PBS to prepare a homogeneous solution, and the samples were kept in the above solution and placed in the incubator. At 0, 1, 5, 15 and 30 h after the experiment, part of the samples were taken out, the excess solution was sucked out and placed in the −80 °C refrigerator to end the degradation experiment, and the samples were freeze-dried together to remove water after the samples in 5 time points were completed, and then the dry weights were recorded and the experimental data were analyzed.

### 2.5. FEF Electrospinning of Unordered GelMA/PEO Microfibers

GelMA/PEO bioink was loaded in a 1mL disposable syringe and a stainless-steel nozzle was assembled. The syringe was covered by aluminum foil to avoid unexpected photo-cross-linking and installed to the fixture of the spinning platform, ensuring that the receiving plate and the spinning head are kept relatively vertical. The distance between nozzle tip and receiving plate was set as 10 mm on the upper computer. The receiving plate was connected to the positive pole of the high-voltage DC power supply and the nozzle was connected to the negative pole. A glass sheet was directly placed on the receiving plate below the nozzle to receive the hydrogel microfibers. After entering the bioink supply speed and amount on the syringe pump controller, bioink supply system was set ON and the high-voltage DC power supply was turned on to start electrospinning. During the experiment, a high-speed camera was used to record the initial formation state of the jet and the morphological changes in the spinning process. The influence of electrospinning parameters on fiber morphology and diameter was explored by changing the PEO concentration, voltage, bioink supply speed.

### 2.6. Crosslinking Process of GelMA/PEO Microfibers

The glass sheet loading the electrospun GelMA/PEO microfibers was infiltrated in aqueous solution containing ammonium sulfate or ethanol. The 405 nm flashlight was placed above the infiltrated glass sheet at a distance of 3 cm and irradiated the electrospun GelMA/PEO microfibers for 1 min to photo-cross-link them.

### 2.7. NEF Direct Writing of Ordered Microfibers

Relative heights of the four angles of the receiving plate were adjusted with the nuts on the four feet to make the receiving plate parallel to the horizontal plane, so as to avoid an uneven microfiber diameter in different areas during the NEF direct writing process. To explore the influence of electrospinning parameters on fiber morphology and diameter in the NEF direct writing, the serpentine straight line routine G-code was applied to control the movement of the three-axis motion platform. The distance between nozzle tips and the upper surface of the glass sheet on the receiving plate was set as 0.3 mm and different voltages, bioink flowrates and nozzle moving speeds were set, respectively. For the direct writing of complex patterns, the bioink flowrate was set as 0.3 μL/min. The voltage was set as 2 kV. Nozzle moving speed was set as 3000 mm/min. The 3D model of the scaffold composed of the NEF direct written ordered microfibers was designed in the upper computer, followed by generating printing routine G-code with the slicing software to control the movement of the three-axis motion platform.

### 2.8. GelMA/PEO Microfiber Scaffold Observation

The crosslinked GelMA/PEO microfiber scaffold was observed or tested by optical microscopy, scanning electron microscopy (SEM) and Fourier infrared spectroscopy. Optical microscopy was used to observe fiber morphology and measure its diameter distribution. SEM was used to observe the microstructure of the fiber surface and the sample was dried and the surface was sprayed with gold for 3 min. Fourier infrared spectroscopy was used to determine the composition of the GelMA/PEO microfiber. PEO powder was dissolved in dichloromethane and coated to potassium bromide salt tablet, followed by tested after it turned into a film with the solvent volatilizing. GelMA powder was dissolved in glacial acetic acid and coated on potassium bromide salt tablet, followed by testing after it turned into a film with the solvent volatilizing. Pure crosslinked GelMA and GelMA/PEO composite microfiber scaffold were frozen in liquid nitrogen for 10 min and quickly ground into powder, followed by further grinding at a ratio of 100 mg of potassium bromide powder per 2 mg of the above powder.

### 2.9. Printing of GelMA/PEO-PCL Composite Scaffolds

PCL mesh scaffold was printed with a fused deposition modeling (FDM) 3D printer with a side length of 16 mm and a spacing of 0.7 mm. The obtained PCL mesh scaffold was soaked in 1 mol/L sodium hydroxide solution for 6 h to remove surface oil, and then placed on the glass sheet for GelMA/PEO microfiber FEF electrospinning for 4 h, so as to obtain a macroscopic ordered and microscopic disordered microfiber scaffold.

### 2.10. Cell Culture on the GelMA/PEO Microfiber Scaffold

HUVECs were selected to be seeded on the GelMA/PEO microfiber scaffold to verify the biocompatibility. The scaffolds were cut into some small sheets so that they could be placed in the wells of the 12-well plate. The cut scaffolds were repeatedly soaked in deionized water 3 times (5 min per time) to remove the residual crosslinking solution components. Then, the cut scaffolds were soaked in alcohol for 1 h, supplemented by UV light treatment, and then cleaned with PBS 3 times (5 min per time). Medium was added into the wells of the 12-well plate and the scaffolds were fully spread with the forceps at the bottom of the well plate, followed by placing them in the incubator overnight and HUVECs (for unordered GelMA/PEO microfiber scaffold) and GFP-HUVECs (for ordered GelMA/PEO microfiber scaffold and GelMA/PEO composite scaffold) were seeded on the treated scaffolds on the next day, respectively. During cell culture, the incubator temperature was set as 37 °C and the CO_2_ concentration was set as 5% (*v*/*v*). The culture medium was changed every three days. For unordered GelMA/PEO microfiber scaffold, cell-seeded scaffolds were tested on the 1st, 3rd and 6th days of culture, including cell viability with Calcein AM/PI kit, cytoskeleton observation with rhodamine-labelled phalloidin and marker protein expression with CD31 immunofluorescence staining. For ordGelMAGelMA/PEO microfiber scaffold, cell morphology on the scaffolds were observed on the 1st, 2nd and 3rd days of culture with confocal fluorescence microsGelMA For GelMA/PEO-PCL composite microfiber scaffold, cell morphology on the scaffolds were observed on the 1st, 2nd and 3rd days of culture with confocal fluorescence microscopy. The detailed operations are listed below:

*Cell culturing:* HUVECs and GFP-HUVECs attached to 75 cm^2^ sterile flasks were cultured in a thermostatic incubator at 37 °C, 5% CO_2_. Medium was refreshed with fresh complete medium every 2 days. Cells were passaged when they proliferated to approximately 90% of the surface area of the flask. Complete medium, PBS buffer and trypsin-EDTA solution (0.25%) were heated to 37 °C through a thermostatic water bath for 30 min in advance. The medium in the flask was discarded and 5 mL PBS buffer was added to remove the fetal bovine blood from the cells. Then, the PBS buffer was discarded and 3 mL trypsin-EDTA solution (0.25%) was added to wet the bottom of the flask, followed by incubation in the incubator at 37 °C, 5% CO_2_ for 3 min to detach the cells from the bottom of the flask. Then, we added 3 mL complete medium and transferred the cell suspension in the flask to a 15 mL sterile centrifuge tube, centrifuging at 1000 rpm for 5 min. Next, the supernatant was removed, and 2 mL complete medium was added to resuspend the cells. Finally, we transferred the 1 mL cell suspension to two new 75 cm^2^ sterile flasks, respectively, and added 15 mL of complete medium for further culturing.

*Live/Dead testing:* We diluted Calcein AM and PI inside the kit with PBS buffer to the concentration of 2 μM and 8 μM, respectively. The prepared Live/Dead staining solution was added to the cell samples, which were then incubated for 40 min at room temperature in the dark. Then, the Live/Dead staining solution was discarded, and PBS buffer was added to the cell samples for washing for 5 min, and repeated 3 times. The stained cell samples were captured by the confocal fluorescence microscope with 488 nm and 561 nm lasers, respectively. Green signals represented living cells and red signals represented dead cells. Finally, the captured images were imported into the ImageJ software for viability calculation (n = 3).

*Rhodamine-labelled phalloidin staining:* The cell samples were washed with PBS buffer for 5 min and repeated 3 times. Then, we added 4% paraformaldehyde to fix the cell samples for 30 min. We removed paraformaldehyde and washed the cell samples with PBS buffer for 5 min, repeated 3 times. Permeabilize cell samples with 0.5% Triton X-100 for 5 min. We removed Triton X-100 and washed the cell samples with PBS buffer for 5 min, repeated 3 times. The phalloidin working solution labeled with 100 nM rhodamine was prepared using PBS buffer containing 1% (*w*/*v*) BSA. We added the phalloidin working solution to the cell samples and incubated them at room temperature in the dark for 30 min. Then, we removed the phalloidin working solution and washed the cell samples with PBS buffer for 5 min, repeated 3 times. We added DAPI solution to the cell samples and incubated them at room temperature in the dark for 5 min. We removed the DAPI solution and washed the cell samples with PBS buffer for 5 min, repeated 3 times. Finally, we added fluorescent anti-quencher agent to the cell samples, which were captured with the confocal fluorescence microscope by 561 nm and 365 nm lasers.

*The instructions for CD31 staining are as follows:* Wash the cell samples with PBS buffer for 5 min, repeated 3 times. Add 4% paraformaldehyde to fix the cell samples for 30 min. Remove paraformaldehyde and wash the cell samples with PBS buffer for 5 min repeated, 3 times. Permeabilize cell samples with 0.5% Triton-X100 for 5 min. Remove Triton-X100 and wash the cell samples with PBS buffer for 5 min, repeated 3 times. Occlusion of cell samples with immunostaining blocking solution was carried out for 2 h. Remove the immunostaining blocking solution and wash the cell samples with PBS buffer for 5 min, repeated 3 times. Then, dilute CD31 reagent in 1:200 ratios by immunofluorescence primary antibody dilution to prepare primary antibody working solution. Add the primary antibody working solution and incubate at room temperature in the dark for 2 h. Remove the primary antibody working solution and wash the cell samples with PBS buffer for 5 min, repeated 3 times. Dilute Alexa Fluor 647 labeled goat anti-rabbit IgG (H+L) reagent at a ratio of 1:500 using immunofluorescent secondary antibody dilution to prepare secondary antibody working solution. Add Alexa Fluor 647 labeled goat anti-rabbit IgG (H+L) secondary antibody solution and incubate at room temperature protected from light for 2 h. Remove the secondary antibody working solution and wash the cell samples with PBS buffer for 5 min, repeated 3 times. Add DAPI solution to the cell samples and incubate at room temperature in the dark for 5 min. Remove the DAPI solution and wash the cell samples with PBS buffer for 5 min repeated, 3 times. Finally, add fluorescent anti-quencher agent to the cell samples, which were captured with the confocal fluorescence microscope by 647 nm and 365 nm lasers.

## 3. Results and Discussion

### 3.1. Component State in the GelMA/PEO Bioink

According to the normalized Fourier infrared absorption spectra of LAP, PEO and GelMA solution raw materials and crosslinked GelMA and GelMA/PEO mixed solutions (Figure 2A), the absorption peak of PEO was mainly concentrated in the telescopic vibration of C-C bond of 2887 cm^−1^ and the telescopic vibration of C-O bond of 1111 cm^−1^. The two most important absorption peaks before GelMA crosslinking were C = O bond telescopic vibration of 1657 cm^−1^ and N-H bond deformation vibration of 1536 cm^−1^. The infrared absorption spectrum of GelMA remained basically stable after crosslinking, but the absorption of amide I band and amide II band was weakened, which was related to the formation of a network structure of carbon-carbon double bonds after crosslinking. Comparing the infrared spectra of GelMA hydrogel alone and GelMA/PEO hydrogels, it could be seen that the main difference between the latter and the former was at the two peak segments of PEO and the rest were basically unchanged, indicating that GelMA and PEO would not undergo obvious chemical reactions during the electrospinning process and, namely simple physical mixing.

### 3.2. FEF Electrospinning State around Nozzle

The main aim of FEF electrospinning is to apply high-voltage static electricity between the spinning nozzle and the receiving plate at a relatively large distance to generate unordered microfiber scaffolds. The surface of the droplet hanging on the spinning nozzle tip produces charge aggregation under the electric field and gradually deforms, slowly forming a cone-shaped droplet, which is called the Taylor cone [36,37]. When the electric field force overcomes obstructive factors such as the surface tension of the droplet, a charged jet is formed, which stretches rapidly in the air, and finally deposits microfibers on the receiving plate. As a manufacturing technology of micro-nano fibers, electrospinning has the advantages of wide applicability of materials, simple and efficient equipment, and has attracted the attention of researchers at home and abroad.

The FEF electrospinning process was recorded by a high-speed camera, and there was no obvious Taylor cone formation in the initial state during the experiment, which might be related to the low viscosity of the solution during the electrospinning process of the aqueous solution. It could be seen that the jet was excited during the spinning process and first had a relatively straight trajectory, and then bended and spiraled around the nozzle. Under different process parameters, there were three main spinning states in the spinning process, which we named as overhang accumulation state, convex equilibrium state and horizontal liquid surface state (Figure 2B). The spinning state of the GelMA/PEO bioink with different voltage and bioink flowrate was summarized (Figure 2C). The main factors affecting the three states were the bioink flowrate and the loading voltage. Overhang accumulation state generally appeared in the case of small voltage, when the electric field force was too much less than the surface tension of the bioink. The jet was more difficult to form, and the spinning process was easy to interrupt. Moreover, the jet movement speed was slow, so the bioink flowrate was faster than the amount of bioink consumed by the jet, and large droplets gradually accumulated around the nozzle and eventually fell, which should be avoided during the spinning process. In the case of horizontal liquid surface state, the electric field force was too large, which could easily exceed the surface tension to form a jet, the spinning process was relatively smooth, and the bioink supplied to the nozzle tip could be consumed in time. The convex equilibrium state was between the overhang accumulation state and the horizontal liquid surface state and could be regarded as a transition state between the two states. The voltage required to form the convex equilibrium state gradually increased as the bioink flowrate increased. Furthermore, in this state, the liquid level at the nozzle was convex rather than horizontal but unlike the overhang accumulation state. The liquid level could remain in its original state (small convex droplet) even if the spinning duration was maintained for several hours.

### 3.3. Crosslinking Process of GelMA/PEO Microfibers

In the electrospinning process, the water in the bioink would evaporate rapidly during the spinning process and the spinning duration often lasts for a long time, so the electrospun microfibers before crosslinking obtained by the above process are basically in a dry state. On the one hand, the crosslinking reaction requires a liquid environment. On the other hand, however, if the obtained microfiber is directly placed in an aqueous solution, because GelMA has good solubility in water before crosslinking and the fiber diameter is too small, the solubility would be greatly enhanced, and the microfibers would be rapidly dissolved. Thus, how to solve this contradiction becomes the key to obtaining microfiber scaffolds in hydrogel state. Considering that GelMA has the basic properties of most proteins as a gelatin-modified product, namely the low solubility of proteins in high-concentration salt solutions or ethanol, this kind of liquid crosslinking environment could be constructed with low solubility of GelMA in it, which would ensure the GelMA/PEO microfibers wetting for crosslinking.

Therefore, we selected ammonium sulfate and ethanol as the crosslinking environment to crosslink the GelMA/PEO microfibers after electrospinning The crosslinking conditions were set as irradiating with 405 nm flashlight for 1 min at a distance of 3 cm from the upper surface of the receiving glass sheet (Appendix A). The ammonium sulfate solution displayed better crosslinking result, with which the overall shape and diameter of the microfibers did not change significantly under the optical microscope before and after crosslinking (Figure 2D). Thus, we selected ammonium sulfate semi-saturated solution system to crosslink GelMA/PEO microfibers. Further observation of microfiber morphology before and after crosslinking was carried out under SEM microscopy (Figure 2E). The results showed that after crosslinking, the microfiber diameter basically did not change significantly but the surface morphology became rougher with defects and porous structure. One possible reason was that in the dry-wet crosslinking process, due to the aqueous environment, GelMA and PEO were partially dissolved before the crosslinking reaction was completed. Moreover, the PEO could be dissolved after the crosslinking was completed, which had been pointed out that because PEO owns good solubility in water, PEO can be removed by soaking in aqueous solution after mixed spinning, resulting in improving the porosity of electrospinning fiber [38,39].

### 3.4. Parameter Effect on the FEF Electrospun Microfibers

As described above, the effect of PEO concentration, bioink flowrate and the applied voltage on the FEF electrospun microfiber diameter distribution (n = 50) was tested.

In terms of PEO concentration, under the condition of 0.3 μL/min bioink flowrate and 4 kV voltage (Figure 3A). The FEF electrospinning process was carried out for 15 min. It could be seen that at 1% (*w*/*v*) PEO concentration, the diameter of FEF electrospinning microfiber was relatively uneven, and there were some beaded structures. When the PEO concentration is 1.5% (*w*/*v*), the fiber morphology was good, and the diameter distribution is uniform. When the PEO concentration was further increased to 2.5% (*w*/*v*), the number of fibers decreased significantly. The diameter difference among the fibers was large, and irregular bifurcated fibers were produced. The diameters of the microfibers were further measured (Figure 3D). The fiber diameters were basically about 5–10 μm and displayed the characteristics of first decreasing and then increasing with the increasing of PEO concentration. It could be because when the PEO concentration was too low, because the spinning performance of the GelMA/PEO bioink is mainly based on PEO, the spinning performance of the solution would be greatly reduced, and the spinning would not be stable. When the PEO concentration was too high, the reason for the reduction of spinning performance was that the high viscosity bioink was difficult to spray out from the nozzle to form a continuous jet, so the number of spinnings was small, and the fiber distribution was uneven.

In terms of bioink flowrate, the PEO concentration was set at 1.5% (*w*/*v*) and the voltage was 4kV (Figure 3B). The FEF electrospinning process was carried out for 15 min with different bioink flowrates. When the bioink flowrate was low, not only the fiber morphology was poor, but also the diameter distribution was uneven, and the number of fibers was small. When the bioink flowrate was arrived at 0.6 μL/min, the fiber quality was better. When the bioink flowrate continued to rise, it would gradually be faster than the speed of the jet consuming and the spinning state switched from the convex equilibrium state to the overhang accumulation state. The fiber diameter distribution range expanded due to insufficient solvent volatilization and the probability of beaded structure increased (Figure 3E). The statistical results of fiber diameter demonstrated that with the increase of bioink flowrate, the fiber diameter showed an upward trend in the early stage, but when the bioink flowrate reached a certain critical point (0.7 μL/min), the fiber diameter began to fall with the overhang accumulation state appeared as discussed above.

In terms of voltage, the PEO concentration was set as 1.5% (*w*/*v*) and the bioink flowrate was set as 0.3 μL/min (Figure 3C). The FEF electrospinning process was carried out for 15 min. The voltage would directly affect the strength of the electric field. When the voltage was low, the electric field force was not enough to break through the surface tension of bioink around the nozzle tip, resulting in unsmooth fibers. When the voltage gradually increased, the fiber morphology was improved but when the voltage was too large, the jet began to become unstable. When the voltage reached 7 kV, droplets were generated, and beaded structures were greatly increased. By analyzing the statistical results of fiber diameter (Figure 3F), it could be found that with the increase of voltage, the fiber diameter increased steadily, and the diameter distribution range also expanded, which was probably because the jet flowrate increased in the high-voltage electric field and the ability of the bioink to overcome the surface tension was significantly enhanced.

Using orthogonal tests, the influence of the above three factors on fiber diameter was summarized (Figure 3G). It could be found that when only a single condition was changed, the increase in fiber diameter was often accompanied by a decrease in homogeneity. In the selected range, in general, increasing the voltage, solution concentration and bioink flowrate could all increase the fiber diameter. In order to ensure the quality of FEF electrospun microfibers, in the follow-up experiment, we selected a concentration of 1.5% (*w*/*v*), a bioink flowrate of 0.3 μL/min and a voltage of 4 kV as the FEF electrospinning conditions.

### 3.5. NEF Direct Writing of Ordered Microfibers

In order to improve the controllability of fiber deposition and fiber structure, NEF direct writing technology was developed on the basis of traditional electrospinning. NEF direct writing technology can continuously and controllably deposit microfibers on a specific substrate sequentially, with the characteristics of non-contact, repeatable processing, high precision, and good compatibility with organic materials, which can meet the growing demand for large-area micro-nano manufacturing. As an emerging manufacturing process, NEF direct writing technology has been widely used in the fields of tissue engineering scaffolds and flexible electronics.

The biggest advantage of NEF direct writing compared with FEF electrospinning is that it can control the precise positioning and deposition of fibers, so compared with classical disordered spinning, in addition to fiber diameter uniformity and surface morphology exploration, it is also necessary to pay attention to the fiber deposition positioning when evaluating the spinning effect. Two typical printing defects were obtained during the NEF direct writing process, namely twisted fiber paths and uneven fiber diameter (Figure 4A). The former mainly occurs when the distance between the nozzle tip and the upper surface of the glass sheet is too large and the nozzle moving speed is too fast. The latter occurs for a variety of reasons, including low voltage, mismatch between bioink flowrate and nozzle moving speed and uneven platform adjustment. Here, the distance between the upper surface of the glass sheet and the nozzle tip was set at 0.3 mm and the orthogonal tests to determine the effect of different bioink flowrate, voltage and nozzle movement speed on the fiber diameter (n = 50) in NEF direct writing (Figure 4B). The results demonstrated that with the increase of voltage and bioink flowrate, the fiber diameter basically showed an upward trend, but in the process of increasing the liquid supply speed from 0.3 μL/min to 0.5 μL/min, the fiber diameter uniformity decreased significantly, which was mainly because the jet speed did not match the bioink flowrate. Furthermore, with the increase of nozzle moving speed, the diameter of the fiber decreased resulted from the accompanied stretching. Since the speed had the characteristics of real-time response and this variable could be easily controlled in G-code, the fiber diameter could be adjusted in real time by changing the nozzle movement speed for variable diameter printing. In addition, to examine the feasibility of establishing a complicated pattern based on HEF direct writing, the G-code of a variety of geometric patterns was obtained by the pattern generation software (Figure 4C). The bioink flowrate was 0.3 μL/min. The voltage was 2 kV and the nozzle moving speed is 3000 mm/min. The results showed that the established electrospinning system and GelMA/PEO bioink could meet the needs of high-precision printing, which could not only print straight lines and regular geometry, but also build complex and dense geometric patterns.

### 3.6. Cell Culture on the GelMA/PEO Microfiber Scaffold

Gelatin is produced by high-temperature denaturation of mammalian collagen and can be extracted from mammalian bones, tendons or skin by acidic or alkaline hydrolysis. Therefore, gelatin-based prepolymer, such as GelMA applied in this work, has excellent biocompatibility and low antigenicity, and its arginine-glycine-aspartate (RGD) sequence and matrix metalloproteinase (MMP) target sequence remain on its molecular chain, which facilitates cell adhesion, stretching, migration and degradation modification in the 3D environment established by it.

To examine the biocompatibility of the scaffolds based on the unordered microfibers and ordered microfibers, which were electrospun by FEF electrospinning and NEF direct writing, respectively, HUVECs (or GFP-HUVECs) were seeded on these scaffolds. In terms of the scaffolds with unordered microfibers, the results showed that the viability could reach 99% after 6-day culturing (Figure 5A and Appendix A). From the cytoskeleton morphology images (Figure 5B), it could be found that on the first day of culture, HUVECs had successfully attached to the scaffold surface. On the third day, they displayed further spreading. On the sixth day, the cells had formed dense cell groups on the scaffold. Furthermore, CD31, also named as platelet endothelial cell adhesion molecule-1 (PECAM-1/CD31), is always used as a marker protein verifying the existence of endothelial cells. The CD31 staining results verified that this scaffold could maintain the capability of express marker protein (Figure 5C). In terms of scaffolds with ordered microfibers obtained by NEF direct writing, the seeded GFP-HUVECs also displayed obvious spreading (Figure 5D). All of these results verified that the GelMA/PEO bioink and the electrospinning method proposed in this work owned promising biocompatibility.

### 3.7. Printing of GelMA/PEO-PCL Composite Scaffold

As a carrier of cell adhesion and providing cell growth environment that mimics the extracellular matrix in vivo, tissue engineering scaffolds have a wide range of roles in the biomedical field. The ideal tissue engineering scaffold needs to have both good biocompatibility and sufficient mechanical strength. However, manufacturing tissue engineering scaffolds that can simultaneously meet the needs of the cell growth environment and sufficient mechanical strength remains a big challenge. At present, one of the most effective methods to solve this problem is to coat hydrogels with excellent biocompatibility on the substrate with high mechanical strength. Therefore, to further explore the application of the electrospinning process, FEF electrospinning was combined with traditional scaffold 3D printing method, namely FDM printing. PCL mesh scaffold was first printed with a FDM printer and transferred to the upper surface of the glass sheet. The GelMA/PEO microfiber scaffold was directly spun on the PCL mesh scaffold (Figure 6A). The results showed that the macroscopic ordered and microscopic disordered microfiber scaffold could be successfully established (Figure 6B,C). Moreover, GFP-HUVECs were seeded on this composite scaffold (Figure 6D). During the 3-day culturing, it could be found that the seeded HUVECs could smoothly attach to the composite scaffold due to the excellent FEF unordered microfiber scaffold and gradually form directed growth along with the ordered PCL microfibers, indicating the feasibility of the combination of the proposed electrospinning method with traditional printing strategies.

## 4. Conclusions

In this study, based on the GelMA/PEO bioink, the unordered/ordered microfiber scaffolds with precise geometry and biocompatibility were successfully electrospun. Hydrogel electrospinning platform was established to realize FEF electrospinning for unordered microfiber scaffolds and NEF direct writing for ordered microfiber scaffolds. FEF electrospinning process of GelMA/PEO bioink was researched and the effect of the parameters on the electrospinning state and fiber diameters including bioink flowrate, voltage and PEO concentration were explored. Furthermore, based on G-code and NEF direct writing technology, microfiber scaffolds with complicated patterns were also successfully printed and the parameter effect on the direct writing was analyzed. The cell experiment results verified the biocompatibility of the GelMA/PEO bioink and the electrospinning process. Moreover, the GelMA/PEO microfiber scaffold was directly spun on the PCL mesh scaffold printed by fused modeling printing (FDM) method, which could lead to directed cell growth.

There are still many directions to be further explored based on this work. On the one hand, in terms of process, due to the complex mechanism of electrospinning, the spinning process involves a variety of effecting factors. This paper only selects some of them for initial testing. The spinning process would be further studied from the perspective of electric field distribution change and liquid microscopic force, which is expected to improve the effectiveness of the electrospinning process and to provide reliable guidance for the related researchers. On the other hand, in terms of application, the cell experiments in this paper focused on their basic biocompatibility. As an initial exploration, we have tried and realized other applications based on the proposed GelMA/PEO bioink and the electrospinning process, such as establishing capillary model (Appendix A) and single-cell microfibers (Appendix A). In future, we will start with these new application cases to carry out more analysis on the feasibility of introducing these methods into related in vitro or in vivo applications.

As a consequence, we believe that this composite bioink system and the corresponding electrospinning method can effectively solve the problem of hydrogel spinnability and provide a powerful tool for various biomedical engineering processes in the future.

## Figures and Tables

**Figure 1 bioengineering-10-00130-f001:**
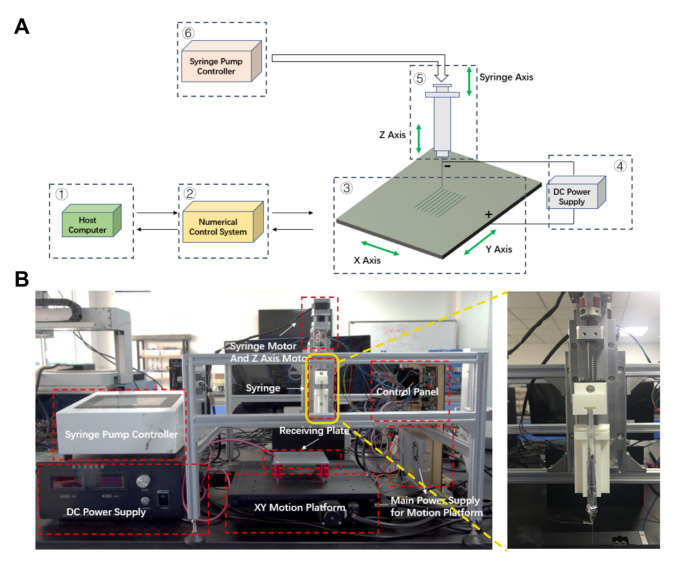
Components of the electrospinning system. (**A**) Schematic diagram of the electrospinning system. (**B**) Image of the electrospinning system and the enlarged image of the spinning/bioink supply module.

**Figure 2 bioengineering-10-00130-f002:**
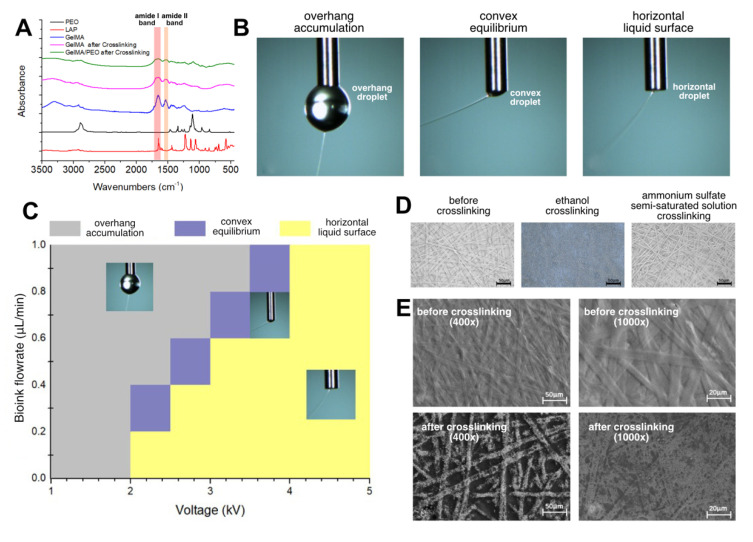
Electrospinning state in the high-voltage electric field and photo-cross-linking process. (**A**) Normalized Fourier infrared absorption spectra of LAP, PEO and GelMA solution raw materials and crosslinked GelMA and GelMA/PEO mixed solutions. (**B**) High-speed camera images of the bioink behavior in the high-voltage electric field. (**C**) Electrospinning state with different parameters. (**D**) Morphology of the crosslinked microfibers in different solution. (**E**) SEM images of the crosslinked unordered microfibers.

**Figure 3 bioengineering-10-00130-f003:**
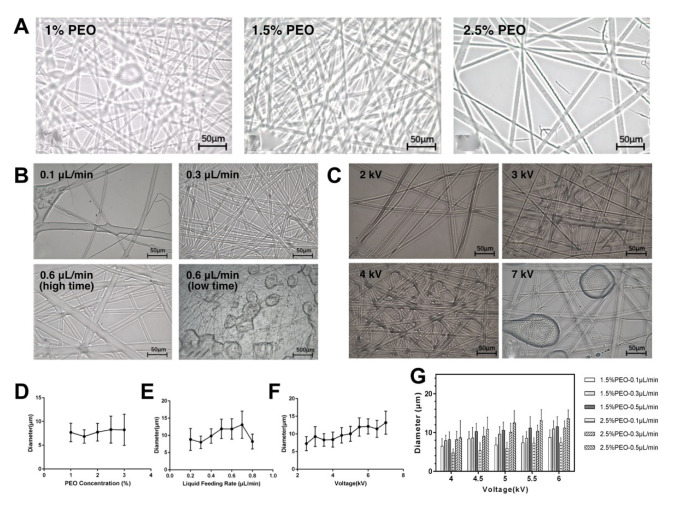
FEF electrospinning analysis with different parameters. (**A**) Unordered microfibers with different PEO concentrations. (**B**) Unordered microfibers with different bioink flowrates. (**C**) Unordered microfibers with different voltages. (**D**) Microfiber diameter distribution with different PEO concentrations. (**E**) Microfiber diameter distribution with different bioink flowrates. (**F**) Microfiber diameter distribution with different voltages. (**G**) Orthogonal tests of different spinning parameters.

**Figure 4 bioengineering-10-00130-f004:**
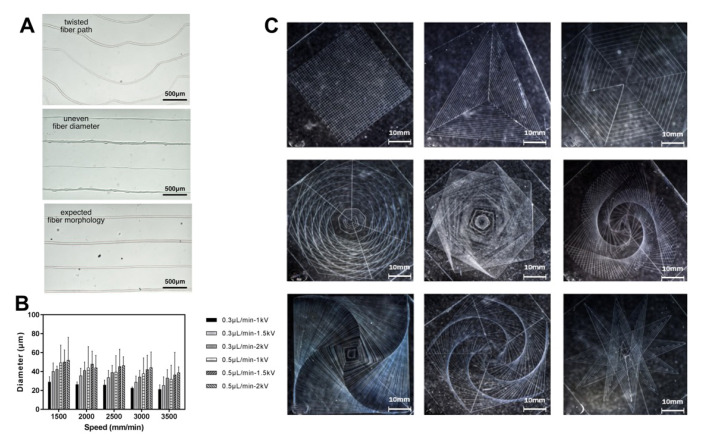
Analysis of NEF direct writing. (**A**) Different microfiber states in the NEF process. (**B**) Orthogonal tests of different spinning parameters. (**C**) Images of the complicated patterns printed by NEF direct writing.

**Figure 5 bioengineering-10-00130-f005:**
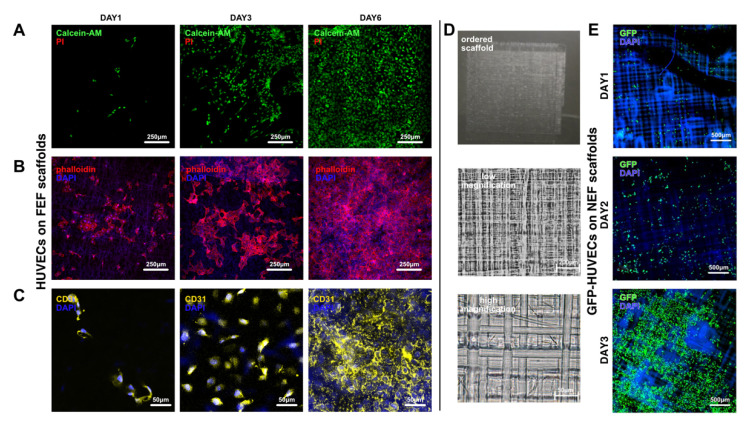
Biocompatibility testing of the FEF/NEF microfiber scaffolds. (**A**) Live/dead testing of HUVECs on the FEF microfiber scaffolds. (**B**) Morphology of HUVECs on the FEF microfiber scaffolds stained by phalloidin and DAPI. (**C**) CD31 expression of HUVECs on the FEF microfiber scaffolds (**D**) Morphology of GFP-HUVECs on the NEF microfiber scaffolds. (**E**) Fluorescence Morphology of GFP-HUVECs on the NEF microfiber scaffolds.

**Figure 6 bioengineering-10-00130-f006:**
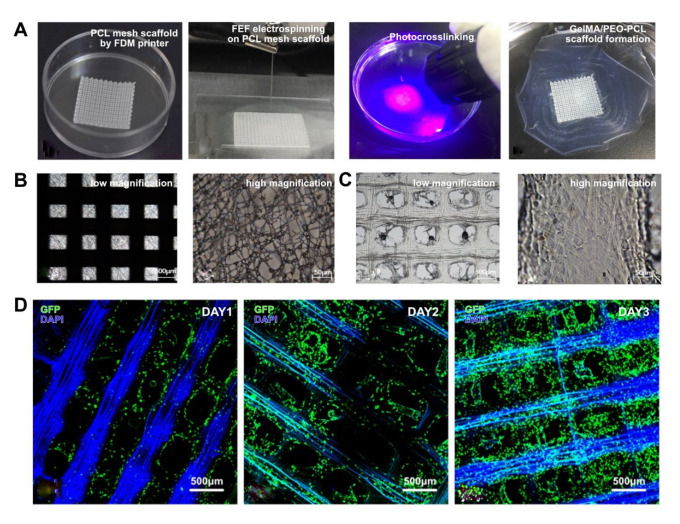
Printing of GelMA/PEO-PCL composite scaffolds. (**A**) Composite scaffold fabrication steps. (**B**) Composite scaffold before crosslinking. (**C**) Composite scaffold after crosslinking. (**D**) GFP-HUVECs morphology on the composite scaffolds.

## Data Availability

The authors declare that all data supporting the findings of this study are available within the paper and its Appendix A or from the corresponding authors upon request.

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
