# Peer review of "Precise Printing of Microfiber Scaffold with Gelatin Methacryloyl (GelMA)/Polyethylene Oxide (PEO) Bioink"

_bioengineering, 2023, doi:10.3390/bioengineering10020130_

Round 1

Reviewer 1 Report

Li et al. designed a three-dimensional (3D) motion platform for electrospinning and built and the spinning process of microfibers under for electric-field (FEF) and near-electric-field (NEF) conditions was systematically studied, respectively. However, the manuscript needs minor revision for publication consideration. My specific comments are detailed below:
1. The title of the manuscript is not clear. Don't use of abbreviation in the title
2.Avoid abbreviations in the abstract.
3. The abstract should include the following points: a summary of your findings; new concepts and innovations demonstrated; a brief restatement of your hypotheses; a comparison with literature-reported results, and possible future work.
4. Provide a brief discussion of the literature-reported studies on the microfiber scaffold in 3D printing. This is important to understand the novelty and significance of this study.
5.  Please assign the characteristic band in the FTIR spectra
6. Results and discussion must be supported with related references.
7. The resolution of figures 3 and 4 is low.
8. In the section "2.3. GelMA/PEO bioink preparation". How did you select the amount of the materials?
9. In the last line of the abstract the authors expressed that this method can effectively solve the problem of hydrogel spinnability and provide a powerful tool for various biomedical engineering in future. How did you conclude this result?. It is better the obtained results compare with other similar work.

Author Response

See in the attached files.

Reviewer 2 Report

This research article introduced a novel method to print disordered and ordered GelMA microfibers by adding PEO material to improve the electrospinning capability, which is difficult for GelMA bioink due to the low electrospinning capability and viscosity. This approach owns high innovative and the data are enough to verified the feasibility of this method. Thus, I recommend this article to the journal.

Minor modifications are listed as below

1. The scale bars in Fig. 2 are not clear enough. I suggest the authors to enlarge the size of the scale bars.

2. Some of the material concentrations need to add (w/v) or (v/v) to avoid the misunderstanding.

3. The introduction of PEO material is not enough. I suggest the authors should add more introduction of PEO material in INTRODUCTION.

4. In Fig. 2B, I suggest the authors should add some symbol and words to highlight the liquid state.

5. The words and grammars are suggested to be further polished.

Author Response

See in the attached files.

Reviewer 3 Report

1.       This is an interesting paper that combines extrusion printing with electrospinning of bio-inks. However, the current work only demonstrated electrospinning of GelMA/PEO bio-ink on the external surface of 3D printed PCL scaffold.

a.       To achieve 3D scaffolds with microscale fiber structures, the authors should demonstrate alternating printing of PCL (extrusion) and GelMA/PEO (electrospinning) instead of only at the surface.

Author Response

See in the attached files.

Reviewer 4 Report

Peer-Review bioengineering – 2150464

The research manuscript entitled “Precise Printing of Microfiber Scaffold with GelMA/PEO Bioink” from Haibing Li et al. is a very interesting study on the development and proof-of-concept of a 3D motion platform for electrospinning able to fabricate GelMA/PEO microfibers under far-electric-field (FEF) and near-electric-field (NEF) conditions. The manuscript fits well within the scope of the journal Bioengineering (ISSN 2306-5354), especially in the Special Issue “Applications of Bioprinting in Medicine”. However, the manuscript has several major issues that must be addressed before being considered for publication.

Issues:

1.     The quality of the written english is poor and must be considerably improved. Throughout the manuscript, there are many sentences that should be rewritten in a clear manner and many verbs in the wrong form. A revision by a native speaker is highly recommended.

2.     Did the authors perform any statistical analysis of the results? This should be included in the manuscript.

3.     The description of the GelMA/PEO degradation assay should be included in the Materials and Methods section.

4.     Figure 2. The caption title needs to be improved. Why not provide SEM images after crosslinking?

5.     How many fibers (and images) were measured to determine the average diameter? This info needs to be included in the manuscript. The authors must also provide histograms with the fiber diameter distributions of all conditions analysed in the manuscript.

6.     Why not show SEM images correspondent to microscopic images of Figure 3? The authors should also provide the respective SEM images.

7.      The graph of Figure 4B is too small and its size should be increased.

8.     Can the authors explain the excessive blue staining in Figure 5D (Day 1)?

9.     The authors must add a quantification assay of cell viability/growth (e.g., AlamarBlue assay, MTT assay or DNA content).

Author Response

See in the attached files.

Round 2

Reviewer 3 Report

The revised manuscript can be accepted in present form

Author Response

Thanks for your approval.

Reviewer 4 Report

The manuscript entitled “Precise Printing of Microfiber Scaffold with GelMA/PEO Bioink” was improved after this round of revisions. 

However, some issues can be improved:

- In comment 6 of the previous revision, the SEM images asked were the ones correspondent to the different conditions of PEO concentration, flow rate, and voltage shown in Figure 3A, B and C.

- Why does Figure S5 not have an error bar? How was the cell viability calculated? From LIVE/DEAD fluorescence microscope images? How many pictures were considered in the analysis? This should be included in the materials and methods section.

- Nevertheless, a more quantitative method (Alamar Blue assay, MTT or DNA Pico green assay) would be valuable.

- Figure 5D. Please correct "DA31" to "Day3".
